# Role of Gut Microbiota and Their Metabolites on Atherosclerosis, Hypertension and Human Blood Platelet Function: A Review

**DOI:** 10.3390/nu13010144

**Published:** 2021-01-03

**Authors:** Asim K. Duttaroy

**Affiliations:** Department of Nutrition, Institute of Basic Medical Sciences, Faculty of Medicine, University of Oslo, 0317 Oslo, Norway; a.k.duttaroy@medisin.uio.no; Tel.: +47-22-82-15-47

**Keywords:** platelet, microbiota, CVD, TMAO, TMA, platelet hyperactivity, GPIIb-IIIa, atherosclerosis, hypertension, polyphenols, short-chain fatty acids

## Abstract

Emerging data have demonstrated a strong association between the gut microbiota and the development of cardiovascular disease (CVD) risk factors such as atherosclerosis, inflammation, obesity, insulin resistance, platelet hyperactivity, and plasma lipid abnormalities. Several studies in humans and animal models have demonstrated an association between gut microbial metabolites such as trimethylamine-*N*-oxide (TMAO), short-chain fatty acids, and bile acid metabolites (amino acid breakdown products) with CVD. Human blood platelets are a critical contributor to the hemostatic process. Besides, these blood cells play a crucial role in developing atherosclerosis and, finally, contribute to cardiac events. Since the TMAO, and other metabolites of the gut microbiota, are asociated with platelet hyperactivity, lipid disorders, and oxidative stress, the diet-gut microbiota interactions have become an important research area in the cardiovascular field. The gut microbiota and their metabolites may be targeted for the therapeutic benefit of CVD from a clinical perspective. This review’s main aim is to highlight the complex interactions between microbiota, their metabolites, and several CVD risk factors.

## 1. Introduction

Cardiovascular disease (CVD) is the single biggest contributor to global mortality [1]. CVD encompasses multiple disorders, including atherosclerosis, hypertension, platelet hyperactivity, stroke, hyperlipidemia, and heart failure [2]. Although genetic and other health conditions are intimately involved, the diet-gut microbiome interactions are increasingly recognized for their contribution to CVD development and progression. Several studies showed an association between gut microbiota and their metabolites with CVD risk factors such as hyperlipidemia, overweight, inflammation, hypertension, and platelet hyperactivity [3], highlighting the intricate relationship between diet, gut microbiota, and CVD [3,4]. In addition to their roles in hemostasis and thrombosis, hyperactive platelets are also important mediators of atherosclerosis. There is strong evidence of platelet hyperactivity in conditions like diabetes, smoking, sedentary lifestyles, aging, obesity, certain gut metabolites, and an unhealthy diet [5,6,7].

Within the human body reside trillions of different microbes, collectively referred to as the human microbiota. The largest microbe population is found in the gut, containing 100 trillion microbes of at least 1000 different bacterial species. Sufficient data indicates that the gut microbiome regulates numerous physiological functions, immune system, cardiovascular system, intestinal function, and absorption and metabolism of nutrients and their metabolites. Several studies have implicated gut dysbiosis in CVD pathology, including atherosclerosis, hypertension, platelet hyperactivity, abnormal lipid metabolism, and vascular dysfunction [8]. Gut dysbiosis is an essential factor responsible for critical CVD risk factors such as atherosclerosis, hypertension, and platelet hyperactivity [9].

Emerging evidence suggests that targeting the gut microbiota and their metabolites can be an effective strategy in the treatment and prevention of CVD [9,10,11]. Numerous metabolites are produced by different gut microbiota species, depending on the diet and the microbiome composition, which affect human health. Among the gut microbiota metabolites, short-chain fatty acids (SCFAs), secondary metabolites of bile acid, and trimethyl-N-oxide (TMAO) are important modulatory factors for various diseases. Plasma levels of TMAO significantly contribute to platelet hyperactivity, abnormal plasma lipids, obesity, and insulin resistance [3,8]. TMAO increases CVD risk factors by altering cholesterol and bile acid metabolism, activating inflammatory pathways, and promoting foam cell formation and platelet hyperactivation, whereas SCFAs contribute to atherosclerosis and hypertension process by different mechanisms. Thus, it is important to investigate cellular signaling involving the gut microbiota metabolites in physiology and pathological states to understand their roles in human health and disease. There is a complex association between gut microbiota, their metabolites, and biological processes affecting CVD risk. Therefore, an in-depth understanding of the roles of the gut microbiota and their metabolites on CVD progression and pathogenesis will help us develop therapeutic potential in the future.

This review describes the current evidence linking gut microbiota and their metabolites with various CVD risk factors.

## 2. Roles of the Gut Microbiota in the Atherosclerosis Process

The atherosclerosis process involves fibrosis of the intima, the formation of fatty plaque, the proliferation of smooth muscle cells, migration of monocytes and T lymphocytes, hyperactive platelets, and cholesterol accumulation [12]. Emerging data suggests that gut dysbiosis can also contribute to atherosclerosis development by increasing systemic inflammation [13,14,15].

Inflammation plays a major role in many diseases, including atherosclerosis [12,16,17]. Accumulated evidence has indicated that gut microbiota and their metabolites play an important role in systemic inflammation and modulate various CVD risk factors [18].

The gut barrier’s integrity is essential for maintaining the host’s health and preventing inflammation and atherosclerosis processes. Intestinal permeability is impaired by the reduced expression of tight junction proteins such as zonula occludens-1, claudin-1, occludin, and creating an imbalance between intestinal epithelial cell death and regeneration [19,20]. *Akkermansia muciniphila* exert protective effects against atherosclerosis by improving gut barrier functions [21]. Gut metagenome analysis showed a relatively lower abundance of *Roseburia* and *Eubacterium,* while *Collinsella* was higher in CVD patients than in healthy individuals [22]. The meta-analysis demonstrated that antibiotic treatment had no significant beneficial effect in CVD [23], even though the gut microbiota plays a vital role in inflammation and CVD risk factors [18]. The integrity of the gut epithelium protects against pathogenic invasion in the systemic circulation, and consequently against immune and inflammatory disorders [24]. When the epithelial integrity is compromised, the invasion of pathogen-associated molecular patterns (PAMPs) leads to an immune response and produces systemic and tissue-specific inflammation. Several PAMPs can stimulate inflammatory processes involving host pattern recognition receptors (PRRs), such as CpG oligodeoxynucleotides flagellin, and lipopeptides [25].

Impaired gut barrier integrity induced by gut dysbiosis is a significant risk factor for chronic inflammation observed in several diseases, including atherosclerosis: the microbial component, lipopolysaccharide, is one of the PAMPs involved in the development of CVD. The association between lipopolysaccharide and CVD risk was first observed in 1999, determined by the endotoxin levels in patients [26], and, later, the association was confirmed by several studies [27,28].

Dysbiosis increases the intestinal permeability by suppressing tight junction proteins, allowing the translocation of lipopolysaccharide into the circulation [29,30]. Gut dysbiosis-derived lipopolysaccharide binds Toll-like receptors (TLRs) and activates downstream immune reaction [31]. Lipopolysaccharide binds TLR4 complex, with its co-receptors cluster of differentiation 14 (CD14). The upregulation of TLRs initiates the inflammation-driven atherosclerosis process [32,33]. The interaction between lipopolysaccharide and TLR4 activates MYD88 and NFκB pathways that lead to an enhanced synthesis of pro-inflammatory cytokines such as IL-6, IL-1, IL-27, and TNF-α. These inflammatory cytokines are involved in atherosclerosis and CVD development [34,35]. Another bacterial PAMP, peptidoglycan, increases CVD risk by disturbing the intestinal epithelial barrier.

Metagenomic sequencing has showed that patients with atherosclerosis had enrichment of genes that encoded peptidoglycan synthesis [22]. The presence of bacterial peptidoglycan was detected in atherosclerotic plaques [36]. The nucleotide-binding oligomerization domain (NOD) proteins, NOD1 and NOD2, drive clearance of the intracellular bacteria or bacterial debris through peptidoglycan recognition by involving NFκB and MAP Kinase [37]. NOD2 is a critical regulator of bacterial immunity and the barrier integrity of the gut. The compositional changes of the gut microflora can modulate CVD risk. Despite numerous data demonstrating pathogenic bacteria’s contribution to the development of CVD, antibiotic trials produced mixed results [38]. A meta-analysis of clinical trials revealed that the modification of gut microbiota by antibiotics failed to demonstrate any benefit regarding mortality due to cardiovascular events in coronary artery disease patients [23].

Furthermore, in an extensive study involving 4012 patients with stable coronary artery disease, the administration of azithromycin showed no effect on cardiac event risk [39]. However, the composition of the microbiota was shown to increase the severity of myocardial infarction in a Dahl S rat model of ischemia/reperfusion injury of the heart. Vancomycin, a poorly absorbable antibiotic, reduced 27% of myocardial infarctions and increased 35% post-ischemic mechanical function recovery [40]. This effect was associated with a change in the gut microbiota (both bacteria and fungi). It reduced plasma leptin, which was later confirmed by administering the leptin-suppressing probiotic Lactobacillus plantarum 299v [40]. These earliest contradictory findings regarding antibiotic utilization (azithromycin vs. vancomycin) explained the complexity of gut microbiota-based intervention in terms of the applied protocol’s efficacy and properties.

Gut microbial metabolites such as methylamines, polyamines, short-chain fatty acids (SCFAs), trimethylamine (TMA), and secondary bile acids play important roles in the host’s physiology and in the development of CVD [41,42]. SCFAs, a group of microbial products (such as propionic acid, acetic acid, and butyric acid) are critically involved in the onset and maintenance of various diseases [43]. A correlation between elevated plasma TMAO levels and atherosclerosis was reported [44,45,46,47]. These microbial metabolites’ involvement in CVD risks in both human and animal models has been extensively reviewed [48].

There is now sufficient evidence that gut microbiota participates in intestinal, cardiovascular health and immune function. The human gut microbiome has predominantly five phyla: *Bacteroidetes*, *Firmicutes*, *Actinobacteria*, *Proteobacteria*, and *Verrucomicrobia* [49]. In the healthy gut, the anaerobic groups *Bacteroidetes* and *Firmicutes* account for more than 90% of the total bacterial species [49]. The gut microbiome has many functions in the host [50]. They are involved in the human digestion of food through two main catabolic saccharolytic and proteolytic catabolic pathways. Both pathways lead to the production of SCFAs from polysaccharides. The second catabolic pathway also produces ammonia, various amines, thiols, phenols, and indoles.

The gut microbiota metabolizes dietary phosphatidylcholine or l-carnitine into TMA [41]. TMA is then taken up the liver and oxidized by hepatic flavin monooxygenase 3 (FMO3), leading to the production of TMAO [8]. One of these is TMA derived from dietary sources of phosphatidylcholine, choline, and L-carnitine. Synthesis of TMAO is secondary to the ingestion of food components containing a TMA moiety, such as choline, phosphatidylcholine, and L-carnitine, all of which are present in high amounts in animal products such as red meat, fish, milk, and eggs. Microbial TMA lyases metabolize these compounds to produce TMA. TMA is then transported to the liver via portal circulation and is oxidized in the liver to TMAO by hepatic FMO, primarily FMO3 [8,51]. TMAO is unique among traditional CVD risk factors in that it is a product of the gut microbial metabolism. The plasma level of TMAO is contributed to by different factors such as diet, gut microflora, drug administration, and liver flavin monooxygenase activity. TMAO enters the systemic circulation and contributes to atherosclerosis development by altering lipid metabolism, platelet activity, obesity, and vascularity (Figure 1). 

TMAO affects platelet activity, lipid metabolism, obesity, insulin resistance, vascular tone, and diabetes, thus stimulating the atherosclerosis process

The use of both prebiotics and probiotics in combination or alone impacts gut microbiota composition. Prebiotics including galacto-oligosaccharides, fructo-oligosaccharides, inulin, etc., stimulate the growth of beneficial microflora, while probiotics have specific beneficial bacterial strains. These interventions can help modulate bacteria to transform precursors into TMA and increase bacteria’s ability to deplete it, or can help modulate bacteria devoid of the genes responsible for converting carnitine or choline to TMA. A majority of bacteria belong to the *methanobacteriales* that reside in the human gut. The methanogenic bacteria have been shown to deplete both TMA and TMAO [52,53]. Resveratrol can also significantly modulate the growth of specific gut microbiota in vivo, including increasing the *Bacteroidetes*-to-*Firmicutes* ratio and the growth of *Bacteroides*, *Lactobacillus*, and *Bifidobacterium* [54,55]. These changes have been shown to reduce the levels of TMAO. Reducing L-carnitine or choline levels in the diet is not a good alternative, as these are important nutrients.

TMAO is associated with obesity, insulin resistance, and renal disease [56]. A link between TMAO and CVD has been established [57]. The circulating levels of TMAO correlated well with atherosclerotic plaque size and cardiovascular events [58]. Several meta-analyses have found the association of the plasma levels of TMAO with CVD and mortality risks [56,59]. Patients with the highest quartile of circulating TMAO level exhibited a higher risk of major adverse cardiovascular events than patients in the lowest quartile [60]. TMAO levels were also associated with vulnerability and plaque formation, long-term risks of cardiovascular events in patients, and poor prognosis [61,62,63].

A high choline diet caused increased TMAO levels and atherosclerosis in animal studies. TMAO mediates, at least in part, the established link between red meat consumption and CVD risk. Therefore, the low blood levels of TMAO as a result of fruits and vegetable intake may account for their cardioprotective effects. Work is ongoing to find novel therapeutic approaches that decrease plasma TMAO levels. The available data indicate modulation of the gut microbial TMAO generating pathway that attenuates atherosclerosis and platelet hyperactivity and in vivo thrombosis potential in animal models; however, the effective treatment modality has yet to be established. Lifestyle modification, including exercise, diet, functional foods, and changing microbiota, could be useful for lowering TMAO levels [64].

TMAO exacerbates the vascular wall’s inflammatory reactions, induces ROS production, and prevents cholesterol reverse transport [65]. TMAO modulated cholesterol and sterol metabolism help to develop atherosclerosis [46]. FMO3 knockdown mice had decreased circulating TMAO levels and attenuated atherosclerosis plaque formation despite activating macrophage reverse cholesterol transport [66,67,68]. The plasma levels of gut microbial dietary phosphatidylcholine metabolites and TMAO produced l-carnitine and γ-butyro-betaine were associated with CVD risk [69,70,71]. The plasma level of TMAO was directly related to atherosclerotic plaque formation [8]. A prospective and observational clinical study of patients with or without chronic heart failure consistently showed that plasma levels of TMAO were positively correlated with the risk of heart failure [72].

The role of TMAO in atherosclerosis was investigated using a dietary choline supplement in ApoE^−/−^ mice. The CD36, steroid receptor RNA activator 1 (SR-A1), and 2 macrophage scavenger receptors were measured. The levels of CD36 and SR-A1 in the macrophages of TMAO-treated mice were increased compared with controls, and antibiotic intervention reduced foam cell formation by decreasing TMA production [8]. No significant impact of TMAO on foam cell formation was observed.

TMAO also promotes atherosclerosis by suppressing reverse cholesterol transport and altering cholesterol transporters’ activity in macrophages [69]. Besides, TMAO suppresses expression of liver bile acid synthetase Cyp7a1 and Cyp27a1 and bile acid transporters, Oatp1, Oatp4, Mrp2, and Ntcp, leading to deranged bile acid-related pathways and promotes atherosclerosis [71]. Farnesoid X receptor (FXR) also controls bile acid metabolism and TMAO production by regulating the expression of hepatic FMO3 [51]. FXR protected mice against atherosclerosis by inhibiting the expression of CYP7A1 and CYP8B1 in ApoE^−/−^ mice [51,66,73,74].

TMAO accelerates atherosclerosis by several mechanisms such as promoting cholesterol influx, inhibiting cholesterol efflux, blocking the bile acid pathway, and causing hyperactivity platelets. TMAO also upregulated the expression of vascular cell adhesion molecule-1 (VCAM-1) and activated protein kinase C (PKC) and NFκB. TMAO thus stimulates atherosclerosis via endothelial cell dysfunction and increases the adhesion of monocytes [75]. These findings have confirmed the roles of TMAO in CVD. However, further work is required to establish how TMAO can be used as a biomarker for CVD risk and atherosclerotic diseases [76,77,78].

TMAO may be regarded as an independent risk factor for CVD. However, inconsistent results were also observed, especially in broad population observations [79,80]. Choline is generally considered a dietary source of TMAO; however, there was no substantial evidence of significant associations between choline intake and CVD risk [81]. Administration l-carnitine resulted in a significant increase in circulating TMAO levels in ApoE(^−/−^ ) mice, but its status was inversely correlated with aortic lesion size [82]. Several population studies in different countries showed that dietary choline and betaine intake was not associated with CVD’s pathogenesis [79,83]. However, more in-depth studies are required for definitive conclusions on the exact roles of TMAO in atherosclerosis, as well as the validation of its therapeutic potential by targeting TMAO-producing bacteria or enzymes.

Another gut microbiota-derived metabolite, bile acid, is involved in various metabolic diseases [84]. Bile acid is stored in the gallbladder and released into the intestine to aid the absorption of dietary lipids and lipid-soluble vitamins. Primary bile acids are usually metabolized by the gut microbiota-derived enzymes into secondary bile acids, including deoxycholic acid, lithocholic acid, hyodeoxy-cholic acid, and ursodeoxy-cholic acid [85]. Suppression of hepatic bile acid biosynthesis may also inhibit high-fat diet-induced gut microbiome alterations, showing the presence of the liver–bile acid–gut microbiome metabolic axis [86]. Recently, the bidirectional relationship between gut microbiota and bile acid metabolism [87] in CVD has been reviewed [48].

Bile acids can accelerate atherosclerosis development through bile-salt hydrolase activity and bile acid receptors [88,89]. Bile-salt hydrolase is present in many bacteria, and archaea such as *Methanobrevibacter smithii*, *Clostridium*, *Enterococcus* [90,91]. Bacteria-mediated bile–salt hydrolase activity can promote atherosclerotic progression by stimulating cholesterol accumulation, foam cell formation, and increasing the atherosclerotic plaque size [92].

TGR5, the G protein-coupled receptor, is an important bile acid receptor of the host that mediates the systemic effects of bile acids [93]. TGR5 can inhibit atherosclerosis development by reducing macrophage-mediated inflammation and lipid loading [94]. Pregnane X receptor (PXR) also regulates the expression of genes involved in the biosynthesis, transport, and metabolism of bile acids. PXR is activated by secondary bile acids [95]. Activation of PXR increases atherogenic lipoproteins VLDL and LDL [96]. Development of atherosclerosis is retarded in PXR and apoE double knockout (PXR^−/−^ and ApoE^−/−^) mice [97].

The microbiota-derived secondary bile acids play essential roles in atherosclerosis development by modulating various bile acid receptors such as FXR, PXR, TGR5, and VDR, and S1PR2. Favorable modulation of bile acid metabolism by targeting microbiota may also prevent the atherosclerosis process [98].

## 3. Contribution of the Gut Microbiota to Hypertension

Hypertension is considered the most preventable risk factor for CVD, and successful hypertension prevention and treatment are key to reducing the risk of CVD [99]. Convincing data on the gut microbiota’s involvement in metabolic diseases [22,91,100,101] has suggested the association between gut microbiota and hypertension [102]. Recently, studies have found an association of gut microbiota with hypertension. Animal studies demonstrated that germ-free rats had elevated blood pressure, underscoring gut microbiota roleta in blood pressure regulation. Antibiotic treatment produced higher blood pressure, which implicated gut microbiota’s probable involvement in regulating blood pressure [102]. In spontaneously hypertensive rats, a significant decrease in microbial richness and diversity was observed with an increased ratio of *Firmicutes*/*Bacteroidetes* [103].

Generally, blood pressure is regulated by the amplitude of vasoconstriction and vasodilation of blood vessels [104]. In spontaneously hypertensive rats, a significant decrease in the composition of microflora in the gut was reported, associated with an increase in the ratio of *Firmicutes*/*Bacteroidetes* [103]. Infusion of angiotensin II(AngII) attenuated the blood pressure increase in germ-free mice compared with conventionally raised mice, indicating that gut microbiota involves blood pressure regulation [105]. The gut microbiota is probably engaged in developing hypertension, though the mechanism is not yet fully elucidated. The gut microbial metabolite, SCFAs, and their effect on ox-LDL levels and other pathways may develop hypertension.

SCFAs such as acetate, propionate, and butyrate play crucial roles in maintaining the gut microbiome’s homeostasis and host immunity [106,107,108]. Interestingly, some gut bacteria but not all produce SCFAs using polysaccharides as substrate [109]. The predominant acetic acid-producing bacteria are *Streptococcus* spp., *Prevotella* spp., *Bifidobacterium* spp., *Clostridium* pp., *A. muciniphila*, etc. [109], whereas propionic acid is produced by *Bacteroides* spp., *Salmonella* spp., *Dialister* spp., *Veillonella* spp., *Roseburiainulinivorans*, *Coprococcus catus*, *Blautia obeum*. [109]. *Lachnospiraceae*, *Ruminococcaceae*, and *Acidaminococcaceae* families mainly produce butyric acid in the gut [110]. The high abundance of butyrate-producing bacteria is associated with lower blood pressure in pregnant women with obesity [111]. Supplementation of fiber and acetate improved gut dysbiosis by increasing the abundance of *Bacteroides acidifaciens*, which was shown to play a beneficial role in hypertension in a mice model [112].

The role of various G-protein-coupled receptors (GPRs) in hypertension was reviewed [113]. The gut microbial metabolites, SCFAs, modulate the activity of GPRs, including GPR41, GPR43, and GPR109A [114]. SCFAs regulate blood pressure by the synthesis of renin in the angiotensin-renin system via GPR-regulated pathways [115,116]. GPR41 knockout mice had high systolic blood pressure compared with wild-type mice, and SCFAs lowered blood pressure by activating endothelial GPR41 [117]. SCFAs produce a hypotensive effect via vasodilation in mice through modulation of olfactory receptor 78 (Olfr78) activity [114,118]. Antibiotic treatment altered gut microbiota composition resulting in increased blood pressure in Olfr78 knockout mice [116]. Overall, these studies demonstrated that gut microbiota might play significant roles in host blood pressure by SCFA-mediated mechanisms. However, the potential for SCFAs as a therapeutic target needs further in-depth investigations.

Dysbiosis can promote oxidation of LDL to oxLDL [119,120]. Thus, gut dysbiosis also contributes to hypertension through vasoconstriction mediated by oxLDL [121]. Higher levels of oxLDL contribute to hypertension by inhibiting nitric oxide synthesis (NO) and endothelin-1 [122]. NO, an important vasodilator, is produced from l-arginine by NO synthase. oxLDL may increase blood pressure by decreasing the production of NO and thus reduces vasodilation [121]. Endothelin-1 plays a crucial role in the maintenance of vascular tension and cardiovascular system homeostasis. Endothelin-1 produces vasodilation at low levels by activating the endothelial receptor B and NO production, but induces vasoconstriction at high levels by increasing oxLDL level via activating the endothelial receptor A [123]. However, the association between gut dysbiosis and hypertension [11,124] requires further study for definitive conclusions. The mechanisms associated with the effects of the gut microbiota in hypertension are depicted in Figure 2.

SCFAs, short-chain fatty acids; GPRs, G-protein-coupled receptors; Olfr78, olfactory receptor 78; NO, nitric oxide; OxLDL, oxidized low-density lipoprotein; ETA, endothelin receptor A. For details, please see the text.

## 4. Platelet Hyperactivity is an Important Contributor to the Development of CVD

Platelets have various functions in physiology. Apart from hemostasis, platelets are involved in several cardiovascular processes, such as atherosclerosis, immune system, inflammation, interaction with other cells, and cardiac events [125,126,127]. Upon activation, platelets secrete hundreds of components from their intracellular stores. Platelet α-granules secrete multiple cytokines, mitogens, and other components that contribute to the CVD processes. In addition to the several membrane receptors described above, activated platelets also express CD62 (P-selectin) and shed membrane particles [7,128].

The platelet membrane microparticles have multiple functions and contribute to thrombus and foam cell formation; they are involved in atherosclerotic processes, blood vessel activation, and inflammation. Thus, inhibiting platelet activation/aggregation can protect against CVD development that affects millions of people worldwide. Though aspirin is still the major anti-platelet therapy, it does not benefit all, as evidenced by the phenomenon of aspirin resistance and unwanted side effects. Therefore, aspirin is not recommended for use in the primary prevention of CVD.

Atherosclerosis is a continuous process promoted by several risk factors, including hypertension, hyperlipidemia, cigarette smoking, diabetes mellitus, dyslipidemia, and platelet hyperactivity. Platelet hyperactivity is a major clinical feature observed in hypertension, diabetes mellitus, obesity, and several other metabolic and vascular diseases [129,130]. Sufficient evidence indicates that hyperactivity of platelets plays an important role in the development of atherosclerosis and the incidence of CVD [129,131]. Blood platelets play a significant role in the development and progression of atherosclerosis [128]. Platelets’ pathophysiologic state is the important underlying CVD risk in diabetes, smoking, obesity, sedentary lifestyle, and other conditions. There is clinical evidence that an increased platelet number, platelet activation, and platelet hyperreactivity are associated with cardiovascular events in acute coronary syndrome patients. Platelets showed increased spontaneous activation in patients with diabetes and hypertension [132,133], promoting thrombus formation. Platelet hyperactivity is associated with the secretion of different components. The shedding of membrane particles plays a vital role in developing atherosclerosis, blood flow, inflammation, immune response, and hypertension. Platelet membrane proteins such as GPIbα, GPV, GPVI, amyloid βA4, TLT-1 (TREM-like transcript-1), P-selectin (CD40L), amyloid-like protein 2, and semaphorin 4D are the most abundantly shed platelet proteins during activation.

Human blood platelets become hyperactive in diabetes mellitus, hyperlipidemia, high fat diet, sedentary lifestyle, etc. The gut microbial metabolite, TMAO, also induces hyperactivity in blood platelets. Hyperactive platelets contribute to atherosclerosis development by different mechanisms. Besides, hyperactive platelets are involved in penultimate thrombotic events. 

Figure 3 describes the effects of different factors, including TMAO, on the development of platelet hyperactivity. Hyperactive blood platelets interact with vessel walls by shedding macro-particles, secreting several adhesive growth factors, and inflammatory agents interrupt the blood flow and produce a pro-thrombotic state in people with obesity, diabetes, a sedentary lifestyle or hypertension, and dysbiosis leads to the production of TMAO. Platelets play an essential role in the development of atherosclerosis and cardiovascular events. Plasma levels of TMAO can induce the hyper-activity in human blood platelets as observed in diabetes and hyperlipidemic conditions. Different conditions, such as insulin resistance, diabetes, sedentary lifestyle and high-fat diet, including TMAO, can induce hyperactivity in blood platelets. Different pathways such as shedding of membrane particles, cytokines, and growth factors activating blood vessels contribute to atherosclerosis, decreased blood flow, and hypertension. In addition to recruiting platelets at the site of the damaged vessel, vWF is involved in atherosclerotic plaque development. P-selectin (CD62P) in platelets stimulates monocytes and macrophages to release chemokines that promote platelet–monocyte aggregates. Activated platelets alter the chemotactic and adhesive properties of endothelial cells by releasing inflammatory molecules. CD40 ligand (CD40L) released from platelets induces inflammatory responses in the endothelium. Several platelet-derived chemokines and growth factors are detectable in atherosclerotic plaques.

Platelet-derived CD40L release pro-inflammatory cytokines from vascular cells in the atheroma to stabilize platelet-rich thrombin and inhibit the re-endothelization of damaged vessels. When activated, platelets release different growth factors such as platelet-derived growth factor (PDGF) and vascular endothelial growth factor (VEGF), membrane particles, and cytokines that participate in atherosclerosis development by promoting vascular smooth muscle cell proliferation [134]. Platelets also secrete 5-hydroxytryptamine, ADP, ATP, and lysophosphatidic acid [134]. Hyperactive platelets produce more reactive oxygen species (ROS), enhancing platelet activity by decreasing NO bioavailability and lowering the intracellular concentration of Ca^2+^ [135]. Furthermore, excessive platelet activation is also attributed to the high mechanical shear forces in the circulation, reduced blood flow, and vascular damage, which is observed in patients with hypertension and diabetes [136]. Platelet hyperactivity plays a causal role in triggering and maintaining the pro-inflammatory and pro-thrombotic state of obesity, creating an environment favorable for atherothrombotic vascular events.

Besides these above pathological conditions, the gut metabolite TMAO also activates circulating blood platelets by elevating Ca^2+^ release from intracellular stores, contributing to increased risk of CVD and plaque formation [3]. TMAO produces IP3 by breaking down platelet membrane phospholipids, thus triggering intracellular Ca^2+^ release from internal stores, leading to platelet activation. In several animal models, acute elevation of circulating TMAO by infusion was shown to enhance in vivo thrombosis potential [51]. In humans, TMAO increased platelet hyperactivity [62] and altered cholesterol metabolism [137].

Therefore, the maintenance of regular platelet activity is critical to maintaining hemostasis. The most predominant anti-platelet agent, aspirin, has a number of severe side effects, making it unsuitable for the primary prevention of CVD. Moreover, aspirin-treated platelets do not have the potential to be involved in other physiological functions, apart from aggregation, in their lifetime. The use of anti-platelet drugs is not recommended as a primary preventive measure. Therefore, it is imperative to find alternative and safe anti-platelet agents to tame hyperactive platelets to reduce the risk of CVD development. Reducing platelet hyperactivity by TMAO may have a significant impact on atherosclerosis and thrombotic complications. Thus, a different approach to suppressing platelet function is needed. However, as of yet only a limited number of food-derived compounds have been investigated in depth for their anti-platelet effects clinically. High intakes of fruit and vegetables and moderate intake of marine fish can lower platelet hyperactivity to some extent [138,139]. The most studied reversible anti-platelet regime is derived from tomatoes that contain several polyphenols, nucleoside derivatives, and phenolic compounds [140]. The potent anti-platelet compounds were isolated in water-soluble tomato extract (known as Fruitflow^®^), as this significantly inhibited platelet aggregation both in vivo and ex vivo [140,141,142]. Human volunteer studies demonstrated the potency and bioavailability of active compounds in Fruitflow^®^. Fruitflow^®^ is a functional product approved by the European Food Safety Authority (EFSA) [141]. Fruitflow^®^ can serve as a safe anti-platelet prophylactic treatment for those at high risk or as an alternative to pharmacological compounds with side effects g [140]. The accumulated data indicate that Fruitflow^®^ may be useful in the primary prevention of CVD. Proteomic studies have shown that the tomato-derived compounds in Fruitflow^®^ (nucleosides, derivatives of simple phenolics, and flavonoid glycosides) affect platelet proteins with platelet structure, platelet coagulation, platelet membrane trafficking, platelet secretion, redox system proteins, and HSP70s [142]. The downstream effects of these interactions with the platelet proteome are seen in the suppression of P-selectin expression by the platelet and reduction in integrin αIIb/β3 activation [142]. An array of extensive basic, mechanistic, compositional, and human trials is testimony to its vascular benefit.

## 5. Conclusions

Emerging data suggest a strong relationship between microbiota-derived compounds and an increased risk of CVD. Therefore, it is important to investigate further the roles of diets, microbial production of TMAO and SCFAs, and their cellular signaling to determine their effects on cardiovascular physiology. The relationship between platelet hyperactivity and CVD is now established. Given the growing concern over CVD due to platelet hyperactivity as observed in obesity, abnormal lipid metabolism, sedentary lifestyle, insulin resistance, TMAO and many others, a new therapy of probiotics and prebiotics may constitute a suitable primary prevention regime without overly reducing the nutritionally important intake of precursors of TMA, such as choline, betaine, and L-carnitine. Although several drugs are available to treat CVD, it is currently the leading cause of death worldwide.

Sufficient convincing data have emerged on the relationship between gut dysbiosis and CVD. However, further work is required for the establishment of gut microbiota-targeted therapy for CVD. Since various experimental and clinical data on the mechanisms of gut microbiota mediated-development of CVD are available, there is a strong possibility of finding new approaches to treat or prevent CVD. SCFAs and some types of bile acids, or reducing the microbial metabolite, TMA, can be modulated by diets, prebiotics, and probiotics and with specific TMA inhibitors. The gut microbial composition may also be favorably modified with probiotics, prebiotics, and natural components. To this end, well-designed large-scale clinical studies are required to validate the preclinical and other small-sized human trials data. The currently available data allows us to target the gut microbiota and its metabolites to understand CVD mechanisms and develop novel preventative or therapeutic regimes.

## Figures and Tables

**Figure 1 nutrients-13-00144-f001:**
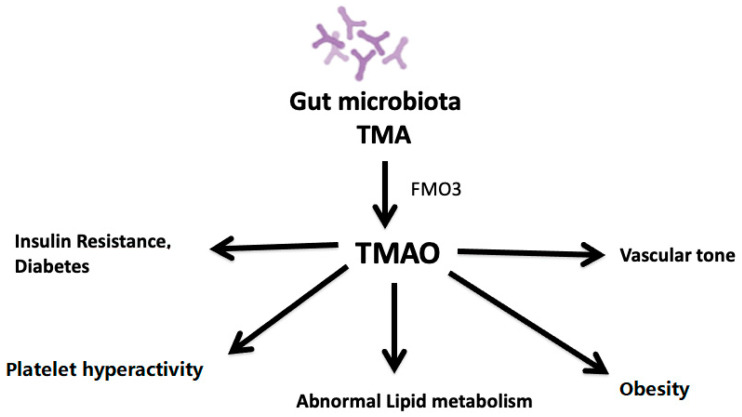
Impact of plasma levels of TMAO on atherosclerosis.

**Figure 2 nutrients-13-00144-f002:**
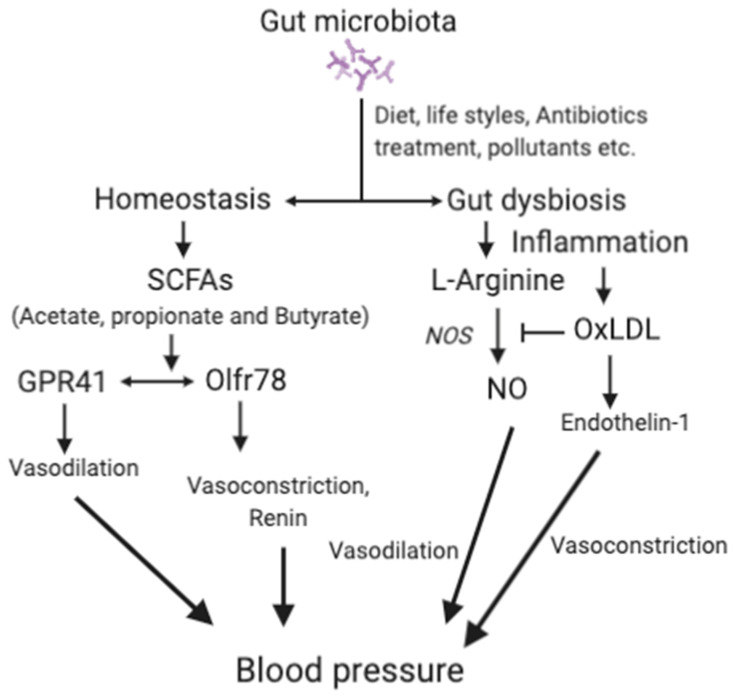
The main mechanisms associated with gut microbiota and hypertension.

**Figure 3 nutrients-13-00144-f003:**
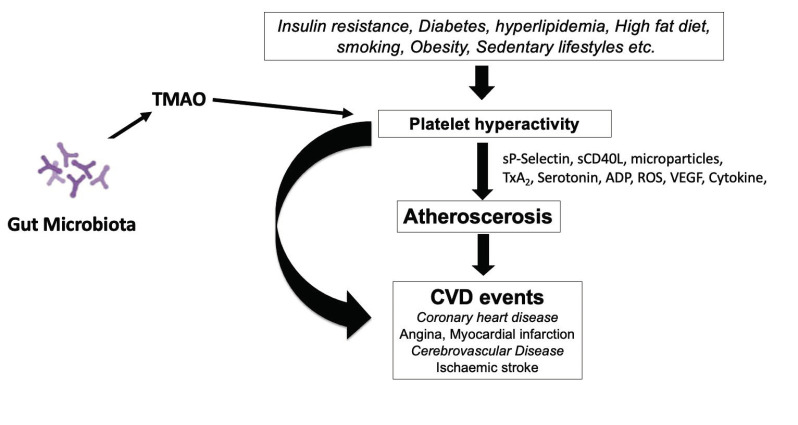
Role of platelets in the development of cardiovascular disease (CVD).

## Data Availability

Not Applicable.

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
