# Peer review of "Role of Gut Microbiota and Their Metabolites on Atherosclerosis, Hypertension and Human Blood Platelet Function: A Review"

_nutrients, 2021, doi:10.3390/nu13010144_

Round 1

Reviewer 1 Report

In my opinion, this review does not add anything new. Throughout thetext the author mixes, at times making it difficult to understand, CVD and atherosclerosis. In the case of hypertension, He refers to LDL, this being perhaps more important in atherosclerosis than in hypertension. The roles of TMAO and TMA have already been deeply reviewed in previous papers and the role of other more recent metabolites such as ImP in diabetes (33208748; 32783890) is not reviewed.

Author Response

Many thanks for allowing me to resubmit the review manuscript entitled “Role of gut microbiota and their metabolites on atherosclerosis, hypertension, and human blood platelet function: A review.

I am grateful to all 4 reviewers for their valuable suggestions, comments, and compliments. I have now addressed all these issues raised by them, and I hope that the revised manuscript is now acceptable for publication in your journal.

#1 Reviewer

In my opinion, this review does not add anything new. Throughout the text, the author mixes, at times at times making it difficult to understand, CVD, and atherosclerosis. In the case of hypertension, He refers to LDL, this being perhaps more important in atherosclerosis than in hypertension. The roles of TMAO and TMA have already been deeply reviewed in previous papers and the role of other more recent metabolites such as ImP in diabetes (33208748; 32783890) is not reviewed.

Reply:

Thanks for the comments. Yes, there are several reviews in this area of research. However, I have included only three risk factors, such as atherosclerosis, hypertension, and hyper-platelet activity. The role of platelet hyperactivity in atherosclerosis development is gaining importance. Hyperactive platelets, in addition to their roles in thrombosis, are also important mediators of atherogenesis. I believe that platelet functionality in relation to TMAO metabolite and atherosclerosis will add an extra value to this review article. This is reflected in the fact that this article's preprint has already been downloaded more than 130 times.

The word “LDL” was mentioned only one time in the whole manuscript to introduce the atherogenic role of OxLDL in the manuscript.

Imidazole-propionate (ImP), the microbial product of histidine, is mostly involved in the pathogenesis of diabetes. Since I did not discuss diabetes in this review, therefore ImP was not mentioned. However, I have now added one line on ImP in the revised manuscript.

Reviewer 2 Report

Author has reviewed the role of metabolites of microbiota on atherosclerosis, hypertension and function of platelets. In particular, he has focused on the role of TMAO. This subject would be of high interest for readership. Role of microbiota has been understadied but during last years its role in different conditions is becoming clear.

The article is quite clear and well written. Author covers the theme broadly and complements text with good figures.

Author Response

#2 Reviewer

Author has reviewed the role of metabolites of microbiota on atherosclerosis, hypertension and function of platelets. In particular, he has focused on the role of TMAO. This subject would be of high interest for readership. Role of microbiota has been understadied but during last years its role in different conditions is becoming clear.

The article is quite clear and well written. Author covers the theme broadly and complements text with good figures.

Reply: Thanks for your positive comments.

Reviewer 3 Report

Generally an interesting and well-done review but there are a few issues to resolve. 

The biggest concern is that there are areas that are indirectly related to the purpose that seem unnecessary, but other areas that are redundant and overlapping even though they are separated in the review.

The area that seems unnecessary is the section from lines 180-232.  the section on HTN summaries the physiology in a few sentences, but here the authors summarize the physiology of platelet activation in an entire page. I feel this is getting too far-off topic and that the authors could get straight to the relationship between the gut microbiota and platelet activation and its role in CVD without this extensive review of basic physiology. 

The areas that seem redundant is the section titled "The role of the gut microbiota in the atherosclerosis process" (lines 68-121) and the section titles "Gut microbial metabolites and their effects on the CVD risk factors" (lines 312-428. The later section seems to cover many of the same topics such as the role of TAMO in atherosclerosis, but in even more depth and detail. My recommendation would be to either combine these two sections or to make them more distinct by focusing the later section on specific pathways not discussed in lines 68-121.

A clear purpose statement is missing.  This should be included for both the abstract and for the main manuscript (in the first paragraph).

There is a typo on line 45 where the authors state there are "trillions of different microbial species...".  I believe they meant to say microbes.

Line 112-113: The statement that antibiotics produce mixed effects need some context and further development. This is an important finding, but the reader is left not knowing any details or if these results are in mice or humans or if there is a preponderance of evidence one way or the other. It is touched on again later but also not developed. 

Line 123-124. Grammatical problem with this sentence.

Table 1:  Please clarify the relationships referred to in this table. For some sections it is clear (the lower 3 rows), but for the upper two rows it isn’t clear how the microbes or metabolites relate to the CVD factors across the top row. For example, is an increase in Lactobacillus associated with atherosclerosis or vice versa? Identify the relationship as is done for the lower rows.

Lines 150-152: This last sentence seems out of place and also has grammatical problems.

Lines 180-183: The text size is smaller in this paragraph.

The section from lines 292-310 seems a little off topic. The last sentence in this section (line 309-310) is not justified. There is one reference for “Fruitflow” and it is a non-systemic review focused on acute thrombosis in Covid-19. This seems like a stretch to say the “accumulated data” when there is in fact little data presented here.

Line 398-400: Not clear what “its status” refers to. Please clarify.

Line 414: Methanobrevibacter smithii is not a bacteria, but a member of the Archaea. Please clarify.

Conclusions:  Line 434-438:  I feel this is an over-statement of the finding from this review and largely based on ref #132 which is more or less an editorial (about the role of nutrition in preventing acute thrombosis in Covid-19) and not a systematic review or clinical trial.

Conclusions:  I feel that the conclusion could do a better job touching on and briefly summarizing each of the sections  

Author Response

#3 Reviewer

Generally an interesting and well-done review but there are a few issues to resolve. 

The biggest concern is that there are areas that are indirectly related to the purpose that seem unnecessary, but other areas that are redundant and overlapping even though they are separated in the review.

The area that seems unnecessary is the section from lines 180-232.  the section on HTN summaries the physiology in a few sentences, but here the authors summarize the physiology of platelet activation in an entire page. I feel this is getting too far-off topic and that the authors could get straight to the relationship between the gut microbiota and platelet activation and its role in CVD without this extensive review of basic physiology. 

Reply: Thanks for your positive comments. This section is now revised extensively. The platelet physiology part is now deleted in the revised manuscript.

The areas that seem redundant is the section titled "The role of the gut microbiota in the atherosclerosis process" (lines 68-121) and the section titles "Gut microbial metabolites and their effects on the CVD risk factors" (lines 312-428. The later section seems to cover many of the same topics such as the role of TAMO in atherosclerosis, but in even more depth and detail. My recommendation would be to either combine these two sections or to make them more distinct by focusing the later section on specific pathways not discussed in lines 68-121.

Reply: Thanks for your comments: All these parts are now combined in the revised manuscript.

A clear purpose statement is missing.  This should be included for both the abstract and for the main manuscript (in the first paragraph).

Reply: Thanks for your comments: The purpose statement is now included in the Abstract section and the first paragraph of the main manuscript.

There is a typo on line 45 where the authors state there are "trillions of different microbial species...".  I believe they meant to say microbes.

Reply: Yes, this is now corrected.

Line 112-113: The statement that antibiotics produce mixed effects need some context and further development. This is an important finding, but the reader is left not knowing any details or if these results are in mice or humans or if there is a preponderance of evidence one way or the other. It is touched on again later but also not developed. 

Reply: This is now well elaborated in the revised manuscript

Line 123-124. Grammatical problem with this sentence.

Reply: This is now corrected in the revised manuscript

Table 1:  Please clarify the relationships referred to in this table. For some sections it is clear (the lower 3 rows), but for the upper two rows it isn’t clear how the microbes or metabolites relate to the CVD factors across the top row. For example, is an increase in Lactobacillus associated with atherosclerosis or vice versa? Identify the relationship as is done for the lower rows

Reply: I agree. The roles of Lactobacillus are complicated in CVD and is difficult to express by arrows in the Table. Similarly, the role of SCFAs in blood pressure regulation is multi-faceted, and complex involves at least two different SCFA receptors, multiple species of bacteria, and numerous host tissues. I think representing their complex actions in simple arrows are complicated and may confuse the readers. Therefore, I have removed the Table in the revised manuscript.

Lines 150-152: This last sentence seems out of place and also has grammatical problems.

Reply:  This sentence has been corrected

Lines 180-183: The text size is smaller in this paragraph.

Reply: This is corrected

The section from lines 292-310 seems a little off topic. The last sentence in this section (line 309-310) is not justified. There is one reference for “Fruitflow” and it is a non-systemic review focused on acute thrombosis in Covid-19. This seems like a stretch to say the “accumulated data” when there is, in fact little data presented here.

Reply: This is now explained and revised, and more references are added now.

Line 398-400: Not clear what “its status” refers to. Please clarify.

Reply: Clarified now in the revised manuscript

Line 414: Methanobrevibacter smithii is not a bacteria, but a member of the Archaea. Please clarify.

Reply: This is now included in the sentence

Conclusions:  Line 434-438:  I feel this is an over-statement of the finding from this review and largely based on ref #132 which is more or less an editorial (about the role of nutrition in preventing acute thrombosis in Covid-19) and not a systematic review or clinical trial.

Reply: This article is a mini-review (Nutr ref #132). I have now added more references in support of the statement.    

Conclusions:  I feel that the conclusion could do a better job touching on and briefly summarizing each of the sections  

Reply: The conclusions section is now revised.

Reviewer 4 Report

In this study entitled “Role of gut microbiota and their metabolites on atherosclerosis, hypertension and human blood platelet function: A review” authors attempted to review the main available data on the association between the gut microbiota and the development of cardiovascular disease (CVD) risk factors, essentially atherosclerosis, inflammation, plasma lipid abnormalities, obesity, insulin resistance and platelet hyperactivity.

They summarized several studies showing a great association has been shown between gut microbial metabolites, such as trimethylamine-N-oxide (TMAO), short-chain fatty acids and bile acid metabolites, amino acid breakdown products with CVD as well. Studies on human blood platelets highlight that these cells are not only essential for the hemostatic process, but also represent a crucial role in the development of atherosclerosis and cardiac events. In particular, some metabolites of the gut microbiota, first of all TMAO, are associated with platelet hyperactivity, condition found in people with diabetes mellitus, sedentary life, obesity, insulin resistance.

This review well describe the main data on the association between gut microbiota, their metabolites and CVD and underline the therapeutic approaches suggested so far.

The study is highly interesting since it deals with the importance of the role of the gut microbiota and the diet-gut microbiota interactions in the development and progression of cardiac events. The review is roughly well written and easy to follow and comprehensive.

However, minor flaws are detected in the manuscript such as:

  • Repetitive information in the lines 49-53, 72-76, 131-132, 181-183, 236-241. It would be better if the same concepts were explained without redundancy
  • In the figure 1, you might specify how NO and endothelin-1 precisely act on blood pressure, following the scheme used for GPR1 and Olfr78 in the same figure
  • Lines 258-260 are part of the description of figure 2, so they should be shortly summarizied and also moved to the entire description only to the caption under the figure
  • It would have been interesting to have further information about the contribution of gut microbiota for every CVD risk factors, not only for atherosclerosis and hypertension, dividing them into appropriate paragraphs

Author Response

#4 Reviewer

In this study entitled “Role of gut microbiota and their metabolites on atherosclerosis, hypertension and human blood platelet function: A review” authors attempted to review the main available data on the association between the gut microbiota and the development of cardiovascular disease (CVD) risk factors, essentially atherosclerosis, inflammation, plasma lipid abnormalities, obesity, insulin resistance and platelet hyperactivity.

They summarized several studies showing a great association has been shown between gut microbial metabolites, such as trimethylamine-N-oxide (TMAO), short-chain fatty acids and bile acid metabolites, amino acid breakdown products with CVD as well. Studies on human blood platelets highlight that these cells are not only essential for the hemostatic process, but also represent a crucial role in the development of atherosclerosis and cardiac events. In particular, some metabolites of the gut microbiota, first of all TMAO, are associated with platelet hyperactivity, condition found in people with diabetes mellitus, sedentary life, obesity, insulin resistance.

 This review well describe the main data on the association between gut microbiota, their metabolites and CVD and underline the therapeutic approaches suggested so far.

The study is highly interesting since it deals with the importance of the role of the gut microbiota and the diet-gut microbiota interactions in the development and progression of cardiac events. The review is roughly well written and easy to follow and comprehensive.

 Reply: Many thanks for the positive comments. I appreciate it very much

However, minor flaws are detected in the manuscript such as:

  • Repetitive information in the lines 49-53, 72-76, 131-132, 181-183, 236-241. It would be better if the same concepts were explained without redundancy

Reply: These sections are now revised.

  • In the figure 1, you might specify how NO and endothelin-1 precisely act on blood pressure, following the scheme used for GPR1 and Olfr78 in the same figure

Reply:  This is now included in the revised Figure-1.

  • Lines 258-260 are part of the description of figure 2, so they should be shortly summarizied and also moved to the entire description only to the caption under the figure:

Reply: This issue is addressed in the revised manuscript.

  • It would have been interesting to have further information about the contribution of gut microbiota for every CVD risk factors, not only for atherosclerosis and hypertension, dividing them into appropriate paragraphs
  • Reply: Thanks for the suggestion. I appreciate it. There are several reviews available on this issue. In this review, I emphasize three important CVD risk factors, including platelet hyperactivity by TMAO,  gut microbial metabolite

Round 2

Reviewer 1 Report

According to the recommendations of the reviewers the author has changed the review in depth. However, I continue thinking that this review is not necessary due to the broad number of reviewS recently publish in this field more detailed than the present one, on the role of gut microbiota in atherosclerosis and atherosclerotic plaque.

https://pubmed.ncbi.nlm.nih.gov/?term=Atherosclerotic+plaque+and+gut+microbiota&filter=pubt.review&filter=datesearch.y_5&sort=date

Sorry, but it is my point of view.